# Tactile-Sensing Technologies: Trends, Challenges and Outlook in Agri-Food Manipulation

**DOI:** 10.3390/s23177362

**Published:** 2023-08-23

**Authors:** Willow Mandil, Vishnu Rajendran, Kiyanoush Nazari, Amir Ghalamzan-Esfahani

**Affiliations:** 1School of Computer Science, University of Lincoln, Lincoln LN6 7TS, UK; 2Lincoln Institute for Agri-Food Technology, University of Lincoln, Lincoln LN6 7TS, UK

**Keywords:** tactile sensing, agri-food, robotics

## Abstract

Tactile sensing plays a pivotal role in achieving precise physical manipulation tasks and extracting vital physical features. This comprehensive review paper presents an in-depth overview of the growing research on tactile-sensing technologies, encompassing state-of-the-art techniques, future prospects, and current limitations. The paper focuses on tactile hardware, algorithmic complexities, and the distinct features offered by each sensor. This paper has a special emphasis on agri-food manipulation and relevant tactile-sensing technologies. It highlights key areas in agri-food manipulation, including robotic harvesting, food item manipulation, and feature evaluation, such as fruit ripeness assessment, along with the emerging field of kitchen robotics. Through this interdisciplinary exploration, we aim to inspire researchers, engineers, and practitioners to harness the power of tactile-sensing technology for transformative advancements in agri-food robotics. By providing a comprehensive understanding of the current landscape and future prospects, this review paper serves as a valuable resource for driving progress in the field of tactile sensing and its application in agri-food systems.

## 1. Introduction

Climate change, shifting migration patterns, urban population growth, ageing populations, and overall population expansion pose significant challenges to the global food chain [1]. The agriculture and food (agri-food) sector can address these challenges through the adoption of robotics and automation technologies. Although the agricultural sector has been slower to integrate automation compared to other industries, there has been a recent surge in agri-food-related robotics and automation research. The inherently complex and uncontrolled nature of the agri-food industry demands innovative solutions for handling uncertainties. Tactile sensation is one such promising approach.

Tactile sensation gives robotics the ability to measure physical interactions and, as such, is developing widespread use in robotics systems [2]. The development of affordable tactile sensors and fabrication techniques has spurred a wide range of research endeavours in tactile sensation within the agri-food sector. From gently harvesting delicate foods without causing damage to assessing the ripeness of soft fruits and sorting produce, tactile sensation equips robots with a novel modality to tackle some of the most intricate physical interaction challenges in the industry.

Tactile sensing in the agri-food sector plays a crucial role in the harvesting, manipulation, and feature extraction of food items. Initial research primarily aimed at utilising tactile sensing for determining key features, such as hardness, in order to estimate food ripeness [3,4,5,6,7,8,9,10,11,12]. This enables the classification of food items, particularly in high-occlusion scenarios, where visual cues may be obstructed by a robot’s end effector, surrounding environment, or packaging [13,14,15,16,17].

Harvesting and grasping food items often involve dealing with visual occlusion and handling delicate, easily bruised items that require a gentle touch [18,19,20,21,22,23,24,25,26,27]. As agricultural robotics progresses towards more complex and realistic environments, these systems must be capable of interacting with their surroundings, such as manipulating foliage during soft fruit harvesting [28,29].

Tactile sensing is also experiencing increased interest in food preparation tasks, such as robotic cooking, where the qualities applied to harvesting and handling are similarly utilised. Tactile sensing allows for measurements without requiring a line of sight, enabling sensing in problems like food item cutting and pouring items from within packaging [30,31,32,33,34,35].

Although tactile sensation is gaining popularity in physical robot interaction tasks, there are still considerable shortcomings with the current approaches. State-of-the-art tactile sensors cannot be spatially scaled up for sensing over large areas due to sensing points crosstalk, wiring space requirements, and latency resulting from having a large number of sensing points [36,37]. While tactile sensation in humans involves an active perception system, where exploratory actions are proactively generated for tactile exploration [38], tactile-based robot controllers are limited to fully reactive systems. Current multi-modal sensing systems and data fusion techniques for effectively combining tactile data with other sensing modalities, such as vision, are still far behind reaching a human-level multi-modal perception. Furthermore, although tactile sensors have been used in robotic systems in agri-food applications, such as robotic harvesting [21,22], fruit ripeness estimation [4,8], and food item classification [13], meeting the standards of food item damage and bruise [39,40,41] or food hygiene standards [42] has not yet been formalised in these research problem statements.

The integration of tactile sensation into the agri-food sector provides benefits in farming yield, product quality control, and food preparation. However, there are significant sectors within the agri-food industry that have not yet utilised tactile sensation. This review paper aims to offer a thorough understanding of tactile sensation and its applications in robotics within the agri-food industry. To achieve this, we first present a general introduction to contemporary tactile sensor hardware and associated algorithms. We then delve into an in-depth analysis of the state-of-the-art techniques and applications of tactile sensation in the agri-food sector. Subsequently, we provide a comprehensive assessment of the limitations of current tactile sensors and their applications, as well as identifying potential future research directions within the agri-food industry. While some topics may overlap with previous surveys, our goal is to equip the reader with a holistic comprehension of this emerging and critical area of research as there is currently no existing review addressing tactile sensation in the agri-food domain.

In this paper, we make the following concrete contributions to the field of tactile sensation research:**Context of tactile sensation research:** We provide a comprehensive overview of tactile sensation technology in Section 2 and core algorithms in Section 3 that are used in robotics and automation research. This discussion offers a solid foundation and understanding of the current state-of-the-art technologies in the field.**Tactile sensation in agri-food:** Our review paper focuses on the application of tactile sensation research specifically in the agri-food domain, which has not been covered in other tactile sensation review papers. In Section 4, we present a concise and comprehensive examination of the current research on tactile sensation applied to various aspects of agri-food, highlighting its significance and potential impact.**Systematic Review of Shortcomings and Challenges:** We contribute a systematic review of the shortcomings and use case challenges associated with the developed tactile sensor technologies for agri-food use cases. This critical assessment, presented in Section 5, addresses an aspect that has been largely disregarded in other review papers on tactile sensors [43,44,45,46,47,48,49].

We outline the structure of this review in Figure 1. By providing a contextual overview, examining the agri-food applications, and addressing the challenges, our paper aims to contribute to the advancement of tactile sensation research and its practical implementation in the agri-food industry.

## 2. Tactile-Sensing Technologies

This section briefly overviews tactile-sensing technologies, emphasising more on the tactile sensors reported in the past five years. This overview includes a discussion of the various transduction methods utilised by these sensors and the tactile features extracted and a discussion of some recent advancements in this domain.

### 2.1. Transduction Methods

Tactile-sensing technologies use various transduction methods to convert physical interactions into useful tactile features. These methods include (a) measuring electric variables (such as resistance, capacitance, impedance, etc.) of the sensing element; (b) analysing sensing element’s deformation using images (camera based), sound (acoustic methods), fluid pressure, etc.; and (c) combining multiple transduction methods. The literature provides detailed reports on how these methods are utilised to derive tactile information [50,51]. However, for the readability of the paper’s subsequent sections, these transduction methods are briefly outlined.

#### 2.1.1. Resistive and Piezoresistive

Both these transduction methods quantify the physical interaction by measuring the changes in resistance of the sensing element (see Figure 2a). In the resistive type, the change in the contact resistance between two conductive elements is transduced as a measure of the applied load [52]. While in the case of the piezoresistive type, piezoresistive materials are used as sensing elements, whose resistance varies with respect to the applied load. The resistive method offers high sensitivity at lower loads and has a good dynamic response. At the same time, it suffers from drift and hysteresis [52]. Piezoresistive methods are known to have high spatial resolution and are less susceptible to noise. But their downside includes lower repeatability, hysteresis, and high power consumption [51].

#### 2.1.2. Capacitive

Capacitive transduction methods measure the changes in the capacitance of the sensing element as a measure of the applied load. The sensing element will have two conductor plates with a dielectric material in between them (see Figure 2b). The capacitive method provides high sensitivity and better spatial resolution with an extensive dynamic range. And the performance is not affected by temperature variations. Due to noise susceptibility and fringing capacitance, capacitive methods call for complex filtering circuitries during the sensor construction [47,51,52].

#### 2.1.3. Magnetic and Hall Effect

The magnetic transduction method utilises the changes in magnetic flux or field intensity when the sensing element is subjected to an external force [47]. These changes in magnetic property are measured using a Hall effect sensing pickup by generating a corresponding voltage relative to the applied load (see Figure 2c). The magnetic method offers better linearity, low hysteresis, and good repeatability in its measurement [53].

#### 2.1.4. Piezoelectric

This transaction method uses piezoelectric material as the sensing element (see Figure 2d). The piezoelectric materials can develop a voltage in response to the applied mechanical load. And this voltage is used as a measure of the applied load. Piezoelectric sensing methods offer a high-frequency response, high sensitivity, and high dynamic sensing range. Also, they are known to have low spatial resolution and are incapable of measuring static loads due to their high internal resistance [51].

#### 2.1.5. Electrical Impedance Tomography

Electrical impedance tomography (EIT) is an imaging technique based on the variation in the electrical impedance distribution on the surface of a deformable object when subjected to an external load [54,55]. Such sensors have two sets of electrodes arranged around a conductive sensing area of the sensor. One set injects an electric current, while the other measures the potential distributions, which vary as a force acts on the sensing area (see Figure 2e).

#### 2.1.6. Camera/Vision Based

In the camera-based method, the image of the deforming sensing surface captured by the camera is used to extract the tactile features (see Figure 2f). Usually, the deforming surface (soft sensing surface) has markers or pins arranged on its inner surface, whose displacements are recorded by the camera [32,56,57,58,59,60,61]. Apart from using pins or markers, the imprints of the external object on the sensing skin are captured by the camera in certain sensors [62,63,64,65,66,67,68]. Compared to other transduction methods discussed above, the camera-based method calls for a bigger form factor of the sensor, as it has to house a camera with its illumination provisions and also needs to place the camera away from the sensing surface to obtain a better view of the surface. To reduce the total form factor of the sensor, attempts have been made to increase the number of cameras so that the gap between the sensing surface and the camera can be reduced without compromising the view of the sensing surface [69].

#### 2.1.7. Optic Fiber Based

With the development of optic fibres, they have been utilised in constructing tactile sensors. Essentially, such a tactile sensor will have a light source (usually LEDs), optic fibre(s), sensing surface, and light output detectors (e.g., camera and CCD) as primary components. The optic fibre acts as the light carrier from the source to the sensing surface and then returns the output light to the detector. The light property variations from any external interaction on the sensing surface are generally quantified either in terms of light intensity modulation, fibre Bragg grating (FBG), or interferometry to output the tactile measurement [47]. This method offers high spatial resolution, sensitivity, and repeatability in the measurements, and the sensor performance is not affected by electromagnetic interference [70].

#### 2.1.8. Acoustics

In the acoustic method, the variation in the acoustic wave propagation through deformable sensing body is mapped to the tactile readings [71,72,73,74,75] (see Figure 2g). The wave propagation alters when there is a deformation on the sensing body due to the external load. This method primarily uses ultrasonic sound waves for the sensor operation.

#### 2.1.9. Fluid Based

An external stimulus can alter the fluid pressure enclosed in a flexible housing. Hence, the measure of this fluid pressure variation can be related to the external stimulus (see Figure 2h). Such a transduction method using pneumatics can generate readings with good linearity, repeatability, and low hysterisis [76].

#### 2.1.10. Triboelectric

This transduction method employs triboelectric nanogenerators (TENGs) to convert the mechanical load into a corresponding electric signal without the need for external operational power [77]. Hence, sensors based on this method are gaining importance, as they can self-power themselves. Triboelectric sensors are an ideal candidate for artificial prosthetics due to their self-powering nature.

#### 2.1.11. Combination of Various Methods

Certain attempts have been reported that use a combination of various transduction methods. Some of them include a combination of the piezoresistive and Hall effect [78]; triboelectric–piezoelectric–pyroelectric [79] and EIT and acoustics [80]. Such attempts offer the advantages of measuring additional tactile features and overcoming the shortcomings of a single transduction method when used alone. For example, combining the triboelectric, piezoelectric, and pyroelectric methods enables the tactile sensor to measure the contact pressure and contact temperature and also makes the sensor operate without an external power [79]. Moreover, combining acoustic with the EIT method enables the sensor to measure contact vibrations, which are impossible with EIT alone [80].

The type of transduction method used in a sensor can define the kind of tactile features it can generate. So, in the next section, various tactile features that these transduction methods can extract are discussed.

**Figure 2 sensors-23-07362-f002:**
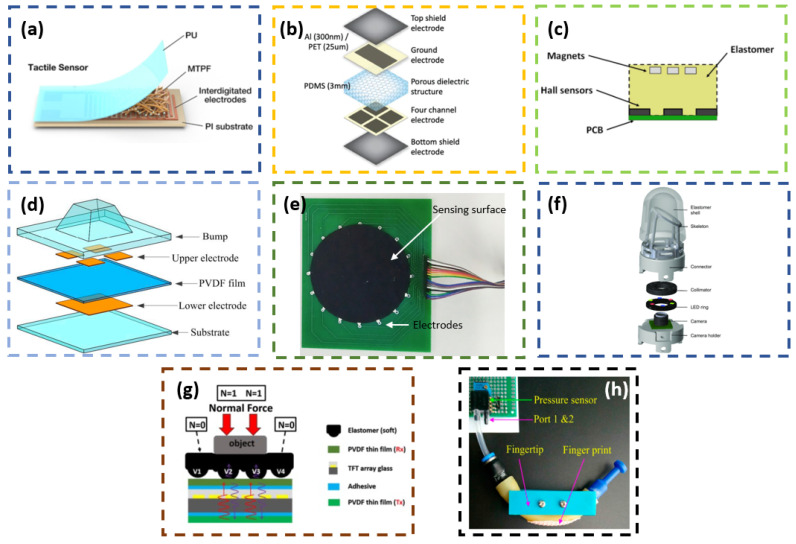
Examples of transduction methods: (**a**) Piezoresistive [81]. (**b**) Capacitive [82]. (**c**) Magnetic [53]. (**d**) Piezoelectric [83]. (**e**) Electrical impedance tomography (EIT) [55]. (**f**) Camera-based [84]. (**g**) Acoustics [71]. (**h**) Fluid based [76].

### 2.2. Tactile Features

Tactile sensing plays a pivotal role in determining the success of numerous robotic manipulation tasks that involve direct physical interactions with the environment [51,85,86]. The tactile sensors help derive essential information for robot control from these interactions using the transduction methods outlined above. In general, this tactile information is collectively known as *“tactile features"*. The tactile features include information on contact force components, contact location, contact deformation, and its derived features (like surface characteristics, the shape of the contacting object, the pose/orientation of contact), and other features like temperature, etc. The details of these features are presented below:

#### 2.2.1. Contact Force

Force-sensing tactile sensors are capable of quantifying the various forms of interaction forces like pressure [55,79,80,81,87,88], normal force [53,71,77,78,89,90,91], shear force [53,78,87,90], tangential force (or frictional force) [89], angular force (or torque) [53,78,87] and vibrations (oscillating force) [76,80]. Measuring contact force is vital for manipulation tasks like grasp force control and slip detection [92].

#### 2.2.2. Contact Location

This is the spatial description of external contact on the sensory surface. It is represented in terms of coordinate values/area relative to the sensor surface. The sensors using transduction methods like EIT [55], magnetic [90], acoustic [91], camera-based [58,63,64], and a few other combinations (EIT-acoustics [80], piezoresisitve–Hall effect [78], triboelectric–piezoelectric–pyroelectric [79]) are reported to generate contact location information.

#### 2.2.3. Contact Deformation

Whenever an external object comes in contact with the deformable sensing surface, the sensing surface takes the shape of the contact object. This deformation can be utilised to derive key features, such as the following: *Contact object shape, surface texture, contact pose, and orientation*. These features help execute tasks like object sorting, mechanical assembly, in-hand manipulation, and preventing object slippage. Since contact deformation is the primary clue to deriving these sub-features, the sensors call for a highly deformable sensing surface. Usually, camera-based transduction methods are used to capture high-definition images of the corresponding deformation of the inner surface of the sensing surface. Using the images of the sensing surface deformation, sensors can reconstruct the shape of the contact object. DenseTact and DIGIT are typical examples of camera-based sensors that can generate contact shape reconstruction from sensing surface deformation. DenseTact can reconstruct shape with an absolute mean error of 0.28 mm [62]. DIGIT is sensitive enough to capture sub-millimetre structures for shape reconstruction [68]. Besides camera-based methods, acoustic and EIT-based transduction methods are also tried to extract the deformation information for contact shape recognition. Ref. [71] showed its capability to differentiate three contact shapes (triangular, square, and circular) while it interacts with the sensing surface at a contact force ranging from 0.2 N to 1 N. The biomimetic elastomeric skin uses the EIT method to recognise the shape of contact objects [80]. The camera-based sensor can also recognise surface texture (e.g., patterns of fingerprints and roughness) of the contact objects from the sensing surface deformation images. Ref. [56] could classify the textures with a maximum accuracy of 83%. Another tactile sensor, NeuroTac, enables surface texture (grid size of 2.5–5 mm) classification at 92.8% accuracy [60]. Tactile sensors like Soft-bubble [67] and OmniTact [66] can use the sensing surface deformation images to estimate the contact pose and sensor orientation. By the contact pose, it means the pose of the object in contact relative to the sensor, and vice versa for sensor orientation. The Soft-bubble tactile sensor [67] can estimate the contact pose in 0.5 s, and the OmniTact sensor [66] can measure the sensor orientation with a maximum median absolute error of 1.986${}^{\circ }$.

#### 2.2.4. Other Features

Apart from all these features, tactile sensors can quantify the temperature of the contacting object. This is made possible by integrating pyroelectric sensing elements in the sensor. One such sensor can measure temperature with a sensitivity of 0.11 V/${}^{\circ }$C [79]. Also, sometimes, measuring the temperature inside actuators (pneumatic fingers) is of interest. For this, acoustic methods are employed, which could sense temperature with a mean accuracy of 4.5 ${}^{\circ }$C [93].

A table is constructed to map the tactile features generated by various transduction methods (see Table 1).

### 2.3. Advancements in Tactile Sensing

Recently, tactile-sensing technology has been receiving updates that can improve their practicality in various robot manipulation tasks. Some relevant ones are briefed here.

#### 2.3.1. Low-Cost Tactile-Sensing Techniques

In most robotic applications, either off-the-shelf or custom-made tactile sensors are attached to the robot counterparts when tactile sensing is needed. This includes attaching sensors to the robot body or to end effectors, depending on the need. In some instances, attaching tactile sensors to the robot counterparts can affect its operation. For example, while attaching a tactile sensor to a pneumatic finger, the flexible nature of the finger can be altered to some extent. So research has been conducted to sensorize end effectors without retrofitting tactile sensors directly. Recently, the acoustic method was tried out in this regard. Zoller et al. [94,95] sensorized a pneumatic finger by implanting a speaker and microphone. The speaker continuously emits a reference sound signal retrieved by the microphone. Whenever this finger interacts with the external environment while in operation, the modulation of the reference sound signal is altered. And this altered modulation is used to characterise the tactile features, like contact force, contact location, the material of a touching object, finger inflation, and temperature [93].

Similarly, acoustic is also adopted to sensorize flexible material skin by making continuous passages between speaker and microphone through the skin [91] (see Figure 3). And any deformation on the skin will impact the cross section of these passages and affect the reference signal modulation. It is proved that such skin can detect normal static forces and their contact location. These approaches make use of minimal hardware and eliminate the requirement of tiny knit electric circuits or other complex manufacturing techniques, thereby reducing the overall cost of the sensor.

#### 2.3.2. Self-Powered Tactile Sensors

Tactile sensors are receiving updates based on their operational power. The vast majority of reported tactile sensors require external power for their operation. Now, a small subset of the tactile sensors can derive power themselves from external interactions. This is achieved by combining transduction elements capable of generating electric charges, such as triboelectric, piezoelectric, and pyroelectric elements [79]. These self-powered tactile sensors possess promising potential for sustained use in applications such as measuring human-specific tactile features in wearable robotic prosthetic applications [77,79].

#### 2.3.3. Anti-Microbial Feature of Tactile Sensors

Tactile sensors are finding application in human health-care purposes, prostheses, robotic surgeries, etc. In such cases, the tactile sensors will make direct contact with the human body, and hence the sensors should exhibit some anti-microbial characteristics for safe usage. There was a recent adaptation in tactile sensors such that they exhibit anti-bacterial features [79,96,97,98,99,100]. This is made possible by using specific materials with anti-microbial properties in the sensor construction, such as fabric triboelectric nanogenerator (FTENG) [100], zinc oxide-polytetrafluoroethylene (ZnO–PTFE) [99], silver nanoparticles (AgNPs) [97], etc.

**Table 1 sensors-23-07362-t001:** Mapping of transduction methods and tactile features extracted.

Tactile Feature	Pr/Re	C	Cr	O	Ma	Pe	EIT	Ac	Tr	F	Com
Normal Force	[101]	[82,89,102,103,104]	[32,56,57,84]	[70,105,106]	[53,90]	-	-	[71,91]	[77]	[76]	[78]
Shear Force	-	[82,87,104]	[84]	-	[53,90]	-	-	-	-	-	[78]
Tangential Force	-	[89,103]	-	-	-	-	-	-	-	-	-
Torque	-	[82,87]	[107]	-	[53]	-	-	-	-	-	[78]
Pressure	[81,88,108,109]	[87]	-	-	-	[110]	[55]	-	[111,112]	-	[79,80]
Vibration	-	-	-	-	-	-	-	-	-	[76]	[80]
Contact Location	[101]	-	[58,63,64,84]	[70,105]	[90]	-	[55]	[75,91,93]	-	-	[78,80]
Deformation/Object Shape/geometry	[101]	-	[59,62,65,68,107,113]	[17,114]	-	-	-	[71,72,74]	-	[115]	[80]
Surface texture	-	-	[60,63,116]	[117]	-	-	-	-	[77]	[76]	-
Pose/Orientation	-	-	[66,67]	-	-	-	-	-	-	-	-
Temperature	-	-	-	-	-	-	-	[93]	-	-	[79]

Pr/Re: Piezoresistive/resistive, C: Capacitive, Cr: Camera based, O: Optic fiber, Ma: Magnetic, Pe: Piezoelectric, Ac: Acoustics, Tr: Triboelectric, F: Fluid based, Com: Combinations.

## 3. Tactile Sensors in Robotics and Automation

Tactile sensing in agri-food robotics has gained attention as an area of interest in the field of advanced agricultural technologies. The focus of this review section is to provide an overview of the current state of the art in tactile sensing for agri-food robotics, discussing the key principles and recent advancements for different agri-food problems. Key challenges and potential future directions are discussed in Section 5 and Section 6, respectively.

Although “Tactile sensing […] has hardly received attention in the agricultural robotics community” (Kootstra et al.) [118], as robotics research drives towards solving real-world agri-food problems, the integration of tactile sensation has become more popular. Typically research has been focused on automated harvesting, where tactile sensation can extract features such as ripeness and location in scenarios where visual sensation typically fails. The following subsections are split into specific agri-food application domains.

### 3.1. Food Item Feature Extraction

Tactile sensation enables the extraction of features that cannot be obtained by visual techniques. Dong et al. [3] showed that objects with similar appearance and shape, e.g., ripe and unripe fruits, “cannot be discriminated accurately with only visual information” and that tactile sensation can be a more effective modality. Tactile sensation is an emerging research area in the domain of food item feature extraction, with a focus on developing robust, non-destructive methods for recognizing the hardness, ripeness, and firmness of various fruits and vegetables.

Zhang et al. [4] focused on recognising the hardness of fruits and vegetables (apple, kiwi, orange and tomato) using tactile array information from the WTS0406-38 magnetic tactile sensor. They proposed PCA-KNN and PCA-SVM classification models, with the latter showing significantly better performance. In a similar vein, Blanes et al. [5] developed a pneumatic robot gripper, shown in Figure 4a, for sorting eggplants by firmness, which demonstrated high sorting accuracy and adaptability. These two studies share a common goal of enhancing the grasping capabilities of robotic manipulators while ensuring product safety. For a bin-sorting application, Ramirez-Amaro et al. [6] used a heuristic-based system to sort soft fruits by ripeness, using learning from demonstration with the group bespoke tactile omni-directional mobile manipulator (TOMM) Dean-Leon et al. [7]; the tactile skin measures torque across the whole body of the manipulator.

Another approach, proposed by Scimeca et al. [8], involves a custom-made gripper with a capacitive tactile sensor array for non-destructive ripeness estimation in mangoes. This method achieved 88% accuracy in ripeness classification, showing potential as an alternative to traditional penetrometers. Ribeiro et al. [9] also explored non-invasive fruit quality control but with a soft tactile sensor that detected small forces and analysed surfaces. This approach demonstrated high-accuracy results for apples (96%) and strawberries (83%), indicating the potential of soft tactile sensors in automated fruit quality control systems. A capacitive sensor was also used by Maharshi et al. [10]. The flexible nano-needle patterned polydimethylsiloxane (PDMS) works as a dielectric layer to detect the ripeness of the fruits.

Cortés et al. [11] introduced a novel robotic gripper, shown in Figure 4b, that combines mechanical and optical properties for non-destructive mango ripeness assessment. This innovative approach demonstrated the potential for improving post-harvest processes by assessing mango quality during pick-and-place operations using data fusion from multiple sensors. Similarly, Blanes et al. [12] presented pneumatic grippers with accelerometers attached to their fingers for assessing the firmness of irregular products, like eggplants and mangoes, showing potential for industrial pick-and-place processes.

The majority of research in tactile food item feature extraction focuses on creating custom end effectors tailored to specific tasks. Researchers employ various sensing technologies, including capacitive, pneumatic, and magnetic systems, often combined with accelerometers and visual sensors. Due to the specialized nature of agri-food research, there is a lack of generalized tools or equipment, resulting in each gripper being designed for a particular task. Although the algorithms used in these studies may be considered generalizable, the hardware is not.

However, Ribeiro et al. [9] and Zhang et al. [4] developed systems capable of non-invasive analysis on multiple different fruits, indicating a potential shift towards more versatile solutions. Encouraging the development of more generalized harvesting end effectors could benefit the sector by producing research that is more widely applicable, allowing researchers to integrate these findings into their specific problem-solving efforts.

### 3.2. Food Item Grasping

The task of grasping and moving food items is the most-studied application domain of tactile sensation in agri-food. Researchers have proposed various methods and designs to tackle the challenges of food item grasping, addressing aspects, such as obstacle interference, damage reduction, and real-time force control.

In recent years, there has been a significant focus on soft robotic grippers for the delicate handling of agricultural products. Cook et al. [18] proposed a 3D-printed tri-gripper with embedded tactile sensors made of thermoplastic polyurethane (TPU), demonstrating a sample application in a fruit pick-and-drop task. Similarly, Liu and Adelson [19] introduced a sensorized soft robotic finger, the GelSight Fin Ray, which passively adapts to the objects it grasps while performing tactile reconstruction and object orientation estimation. This design allows for applications, such as wine glass reorientation and placement in a kitchen task; testing was also shown on soft fruit, highlighting use in agri-food applications as well.

Both Hohimer et al. [20] and Zhou et al. [21] investigated the use of soft robotic actuators with embedded tactile-sensing arrays for agricultural applications. Hohimer et al. [20] explored multi-material fused filament fabrication to print flexible TPU-MWCNT composites with built-in tactile-sensing capabilities for performing apple grasping, while Zhou et al. [21] presented a tactile-enabled robotic grasping method that combined deep learning, tactile sensing, and soft robotics. The work of Zhou et al. [21] is on an intelligent robotic grasping method for handling obstacle interference in crop harvesting environments. The researchers designed a fin-ray gripper with an embedded tactile sensor and multiple degrees of freedom, shown in Figure 5a,b, which can adjust its state based on the tactile feedback. A robust perception algorithm and deep learning network were developed to classify grasping status using stress distribution data from the gripper’s fingers. The method developed is applied to apple harvesting.

Zhou et al. [22] addressed the challenges of robotic harvesting unstructured horticultural environments which contain obstacles, such as branches, twigs, trellis wires, and sprinkler lines. Robotic grasping in these environments can lead to high fruit damage rates (6.3% to 30%). The proposed method, which builds on research in Zhou et al. [21], integrates the fin-ray fingers with embedded tactile-sensing arrays and customised perception algorithms to enhance the robot’s ability to detect and handle branch interference during the harvesting process, thereby reducing potential mechanical fruit damage, shown in Figure 5a,b. Experimental validations demonstrate an overall 83.3–87.0% grasping status detection success rate and a promising interference handling method.

A grasp adaptation controller can adapt its grasp pose to maintain hold of an object; this is especially tricky when dealing with delicate objects, as the typical policy of a large grip force is not acceptable. Yamaguchi and Atkeson [23] used a vision-based tactile sensor that extracts object information, such as distance, location, pose, size, shape, and texture. These features were integrated into a grasp adaptation controller to pick up 30 different and delicate food items, including vegetables, fruit, eggs and mushrooms.

Tactile sensing has also been combined with other sensing modalities to improve robotic fruit picking Dischinger et al. [24]. While previous research had focused on visual feedback for closed-loop end-effector placement, this study aims to incorporate tactile, visual, and force feedback for efficient and damage-free fruit removal from trees. Dischinger et al. [24] presented the design of a custom end effector with multiple in-hand sensors, including tactile sensors on the fingertips. The end effector was tested on a Honeycrisp apple tree in outdoor picking trials, demonstrating the ability to detect fruit slip, separate fruit from the tree, and release fruit from the hand using multi-modal sensing.

Slip detection for fruit grasping and manipulation was also explored in Zhou et al. [25]. Using the tactile sensor presented in Zhou et al. [21], the system uses an LSTM (long short-term memory unit)-based recurrent neural network to process tactile sensation into a slip detection signal for each of the four fingers and a close loop controller to maintain a delicate grasp of apples while under leaf interference.

Tian et al. [26] developed a sensitive slipping sensor with a piezoresistor to control the gripping force of agricultural robots handling fruits and vegetables. By using an adaptive neuro-fuzzy inference system, the researchers were also able to effectively extract the slipping signal and control the gripping force when grasping tomatoes and apples. By analysing the grasp force and slip signals, the system is able to adjust the grip force to reduce bruising from robot manipulation.

Misimi et al. [27] presented a robust learning policy based on learning from demonstration (LfD) for the robotic grasping of compliant food objects. The approach combines visual (RGB-D) images and tactile data to estimate the gripper pose, finger configuration, and forces necessary for effective robotic handling. The proposed LfD learning policy automatically discards inconsistent demonstrations from human teachers and estimates the intended policy. This method is validated for fragile and compliant food objects (tested on lettuce) with complex 3D shapes.

Handling delicate food items like crisps without causing damage was explored in Ishikawa et al. [119]. The system attempted to anticipate fractures in food items during robotic manipulation. The LSTM-based learning system uses a two-fingered end effector equipped with tactile sensing that determines the physical properties of a given food item. A predictive control algorithm is then applied to maximise grip force without fracturing or damaging the food item.

In conclusion, the inherent benefits of tactile sensation can be exploited in grasping delicate food items. Soft robotic grippers with embedded tactile sensors have emerged as a promising approach to delicately handle agricultural products, while the integration of tactile sensing with other modalities has further enhanced efficiency and damage reduction. Research addressing obstacle interference, damage reduction, and real-time force control has enabled the development of intelligent robotic systems that can adapt to various agri-food applications.

### 3.3. Food Item Identification

Fruit identification is a key challenge with some food items during harvesting and crop monitoring. Knowing if the item grasped is the intended target is essential for efficient, timely and safe robotic grasping. Visual-based approaches are often not appropriate due to high occlusion levels during harvesting from the surrounding environment, such as foliage and other crops, and also from the robot’s end effector itself. Tactile sensation is able to operate in areas of low occlusion, both as an independent sensing modality and in collaboration with visual sensation.

Fruit identification in agricultural robotics can be achieved using adaptive robotic grippers, tactile sensing, and machine learning algorithms. Zhang et al. [13] developed a bespoke adaptive gripper, shown in Figure 6a,b, with force and bending sensors to measure contact force distribution and finger deformation during grasping. The random forest classifier demonstrates the highest accuracy of 98% in identifying five fruit types. The proposed method can provide a reference for controlling grasping force and planning robotic motion during the plucking, picking, and harvesting of fruits and vegetables.

Drimus et al. [120] developed an 8×8 tactile sensor for grasping and classifying a variety of soft and deformable food items using a dynamic time warping and a nearest neighbourhood classifier.

A hybrid tactile sensor combining a triboelectric active sensing unit with an electromagnetic inductance transducer developed in Li et al. [14] identified fruits from eight categories with an accuracy of 98.75%. Further, using data from 200 object-gripping trials, they trained a convolutional neural network to process the tactile data. The system was capable of identification of soft fruit through four kinds of fruits wrapped in paper bags, plastic bags, foam and none, with a recognition accuracy of 95.93%.

Riffo et al. [15] took a similar approach, using a pressure sensor and machine learning to categorise soft fruits. Li and Zhu [16] identified objects with a new multi-sensory robot hand. Integrating the tactile features of (i) contact pressure (ii) local ambient temperature, (iii) thermal conductivity and (iv) temperature of an object, together with a neural network classification system, enabled them to distinguish between soft fruits and other items.

Using a VGG neural network on post-processed tactile signals from grasping, Lyu et al. [17] were able to classify soft fruit and vegetables using fibre Bragg grating (FBG) tactile sensors attached to the inside fingers of a three-fingered fin-ray end effector.

### 3.4. Selective Harvesting Motion Planning and Control

Reaching a target fruit for selective robotic harvesting can be a complex task, often requiring more than simple obstacle avoidance. Selective harvesting in greenhouses or orchards often requires robots to cope with dense clutter, non-repetitive tasks, and diverse environmental conditions. Traditional industrial robot systems and vision-based sensors are not well suited for these requirements, as they treat obstacles as rigid bodies and cannot differentiate between soft and hard objects. The ability to manipulate and interact with surrounding obstacles can enable selective harvesting robotics to access more food items when harvesting.

Schuetz et al. [28] proposed a simple and efficient tactile sensor module for a 9-DOF (degree of freedom) multipurpose agricultural manipulator. They developed two approaches for reactive inverse kinematics planning algorithms. By incorporating tactile sensing, the manipulator could perform fine manipulation tasks, explore unknown regions, and respond to the impact of its actions on the surrounding environment structure. The work presents real-world initial experiments that evaluate the performance of the tactile sensor module and the reactive inverse kinematics planning algorithms in suitable agricultural scenarios.

In more recent research, the ability to push aside occluding unripe strawberries was explored in Nazari et al. [29]; Figure 7. Occlusion from a robot’s end effector causes physical interaction tasks, like pushing, to be a significant issue.Tactile sensation can be used to improve pushing performance and a robot’s physical interaction perception Mandil and Ghalamzan-E [121]. Nazari et al. [29] showed that pushing aside occluding strawberries can be performed with tactile sensation alone, enabling harvesting in these complex scenarios.

### 3.5. Food Preparation and Kitchen Robotics

The use of tactile sensation for pouring food items from deformable containers and packaging was introduced in Tsuchiya et al. [30]. Using a dual arm system, coffee beans, rice, flour and breakfast cereal are all poured using tactile sensation (3D force sensors OMD20-SE-40N, On-Robot Ltd. integrated into three fingers of the end effector), which is used to minimise the grasping force and thus the deformation of the deformable containers, thus reducing pour uncertainty.

During the cooking of beef, Wang et al. [31] used pressure reading tactile sensors to measure tenderness. By probing the beef during cooking, the system was able to agree 95% of the time with the high cost and time-consuming established procedure.

Cutting food items in a household robotics setting was explored in Yamaguchi and Atkeson [32]. By combining visual and optical-based tactile sensations, the control system uses force sensations inferred through the knife to avoid slipping and to feel when the knife has cut through the item. Zhang et al. [33] further explored tactile sensation use in food item cutting. The proposed method measures vibration and uses force–torque sensing to control and adapt the cutting motion and to monitor for contact events. The control method was capable of cutting through a variety of different food items, and further, the tactile system could be used to classify food items and make ripeness estimations.

Tactile sensation has been used to identify foreign objects in food items Shimonomura et al. [34]. By rolling a tactile sensor over the item, hardness measures were mapped to the image space and could be identified. Recent advancements towards a generalised dataset and data collection method for kitchen robotics were explored and developed by DelPreto et al. [35]. The method uses human demonstrations with a variety of sensing features for the exploitation of multi-modal kitchen robotics, including tactile sensation from the human demonstrations. This dataset will help integrate tactile sensation into kitchen robotics.

### 3.6. Summary

In conclusion, tactile sensing in agri-food robotics has emerged as a valuable area of research, with applications in food item feature extraction, grasping, identification, selective harvesting motion planning and control, and food preparation. As the integration of tactile sensing becomes more popular, researchers are focusing on developing robust, non-destructive, and adaptable methods for addressing a wide range of agri-food challenges. In the next section, we will present the major shortcomings and challenges of using tactile sensors in real-world applications based on the reviewed research works in Section 2, Section 3 and Section 4.

## 4. Applications of Tactile Sensors in Agri-Food

In this section, we will explore the diverse range of general applications of tactile-sensing technologies, focusing on their essential role in enhancing the capabilities of robotic systems and various other contexts. Tactile sensing has shown promise in numerous applications, including slip control, texture recognition, robot pushing, 3D shape reconstruction, etc. Understanding these general applications is vital, as they provide the foundation upon which more specific agri-food applications can be built. We will present an overview of each application, highlighting the importance of tactile sensing in facilitating better control, interaction, and overall performance in diverse scenarios.

### 4.1. Force Control

Force control is a crucial aspect of robotic systems, as it enables robots to modulate the applied forces during various tasks, such as grasping, manipulation, and assembly. Tactile sensing plays a significant role in force control, as some tactile sensors can provide real-time information about the contact forces and pressure distribution between the robot’s end effector and the interacting object [45]. This information allows robots to maintain stable contact with objects while handling them.

Tactile sensors, such as capacitive, piezoresistive, or optical sensors, are commonly employed in force control applications [86]. They can be integrated into the end effector or gripper, providing valuable feedback about the forces exerted on the object during manipulation [122]. This feedback can then be used to adjust the applied forces dynamically, ensuring safe and efficient object handling. Further, the integration of tactile sensors into multi-fingered robotic hands enables more dexterous force control of objects [45,123,124].

### 4.2. Robotic Grasping

Grasping is an essential component of robotic manipulation, as it involves determining the optimal way to grasp and manipulate objects using a robotic hand or end effector. Tactile sensing plays a crucial role in grasping, as it provides valuable information about the object’s properties, such as shape, size, and surface texture, enabling more effective and stable grasping strategies [125,126].

Tactile sensors can be employed in various stages of robotic grasping. During the pre-grasp phase, contact analysis and force optimisation can be applied to increase the likelihood of a successful grasp [127]. Once contact has been made, these sensors can provide real-time feedback about the contact points and force distribution, allowing the robotic system to dynamically adjust its grasping strategy based on the acquired information [128,129]. Re-grasping, the ability to adapt the grasp pose on the object after the initial grasp, is also an essential aspect aided by tactile sensation. By re-grasping based on learned object dynamics acquired during previous grasp attempts, systems can adjust grasp plans to produce successful tactile grasps [130].

Recent advancements in machine learning techniques, such as deep learning and reinforcement learning, have been utilised in grasp planning to improve grasp quality and stability [131,132]. By incorporating tactile-sensing data, these algorithms can learn more robust and adaptive grasping strategies that can take into account complex geometries and unknown properties like softness and provide more generalised grasp approaches [123]. Multi-modal approaches, like visuo-tactile (a vision and tactile multi-modal approach) methods, have enabled precision grasp planning [133] in the multi-fingered robotic hand, showing the benefits of integrating tactile sensation with visual information for more complex grasp planning.

### 4.3. Slip Detection

Slip detection is a critical aspect of robotic manipulation, as it enables robots to identify and respond to object slippage during grasping and handling tasks. Tactile sensing plays a significant role in slip detection, as it provides valuable information about contact forces, pressure distribution, and object surface properties, which are essential for detecting the onset of slippage and maintaining stable grasps [134,135].

Various types of tactile sensors, such as capacitive, piezoresistive, and optical sensors, have been employed for slip detection in robotic systems [135,136,137,138]. These sensors can detect changes in contact forces or pressure distribution patterns, which may indicate the occurrence of slippage; by monitoring these changes, robots can dynamically adjust their grasping force and strategy to prevent further slippage or to re-establish a stable grasp. Classical methods, like support vector machines and random forest, are still highly used [139,140]. Specifying a threshold on the rate of shear forces for slip detection is a common approach for slip classification [138,141]. Other techniques rely on analysing micro-vibration in the incipient slip phase such as spectral analysis [142,143]. Multi-modal approaches are also proposed to use proximity [144] or visual sensing data [145] to improve slip detection.

Machine learning techniques have been increasingly applied to slip detection, leveraging the rich information provided by tactile sensors [146]. Furthermore, deep learning approaches have shown promise in enhancing slip detection capabilities, particularly when handling objects with complex or unknown properties. Recurrent neural networks, graph neural networks and LSTM (long short-term memory units)-based recurrent neural networks have shown success in this field [123,146,147].

More recent works have attempted to improve robustness, safety and trajectory optimisation through slip prediction [148]. Where tactile sensations are predicted into the future during object manipulation [149], this tactile prediction can then be classified into a predictive slip signal, with which the robot can adapt its path in an optimal manner to avoid future slip [150].

### 4.4. Texture Recognition

Tactile sensing can be used to recognise different textures on an object’s surface. This information can be used by a robot to perform tasks such as sorting objects based on their texture.

Texture recognition is an important capability for robotic systems, as it allows robots to identify and distinguish between different surface properties of objects [151]. Tactile sensing plays a significant role in texture recognition, as it provides rich information about the surface features and properties of objects, such as roughness, hardness, and friction [47].

Various types of tactile sensors, including capacitive, piezoresistive, optical sensors and even tactile features, like surface temperature and vibration, have been employed for texture recognition in robotic systems [60,152,153,154]. These sensors can measure the physical interactions between the robot’s end effector and the object’s surface, allowing the robotic system to extract essential features for texture classification [155].

Although classical methods can be used for texture recognition [60,151,156], machine learning techniques, particularly deep learning, have shown significant promise in enhancing texture recognition capabilities using tactile data [157]. Convolutional neural networks (CNNs) have been used to process and analyse tactile images, enabling accurate and efficient classification of different textures [158,159,160]. More recently, spiking neural networks have also been applied to high frequency and efficient texture classification [161]. As with other tactile sensation problems, visuo-tactile multi-modal approaches can be used for texture recognition to provide extra information for deep learning models to use when performing texture classification [160,162].

### 4.5. Compliance Control

Compliance control is an essential aspect of robotic manipulation, as it allows robots to adapt to the physical properties and geometries of objects, ensuring stable and secure grasping, and minimising damage. Tactile sensing plays a critical role in compliance control, as it provides information about the changing contact forces, pressure distribution, and object surface properties, enabling robots to adjust their grasping and manipulation strategies accordingly [163,164]. Although visual information can be applied to this task, occlusion from the robot’s end effector often makes it impractical to rely on visual information for compliance control.

Various control strategies have been developed to achieve compliance in robotic systems, such as impedance control, force control, and hybrid force/position control [165,166]. These strategies rely on tactile-sensing data to modulate the robot’s stiffness or damping properties, enabling it to adapt to the object’s physical characteristics and maintain stable contact during manipulation [167].

### 4.6. Object Recognition

Tactile sensors have played a crucial role in advancing object recognition capabilities, which refer to the identification and classification of objects based on their physical properties [45]. This is particularly important in unstructured environments, where visual information may be insufficient or unreliable. However, even with unoccluded visual information, tactile sensation features like local surface texture [156] are proven to improve performance in object recognition tasks [86].

Exploiting multi-modal tactile sensors to provide more features improves tactile object recognition [168]. Spectral analysis from sliding a tactile sensor over the surface of the object can also provide the required features for object recognition [169]. Using image-processing techniques applied to contact pattern recognition can also be used to recognise objects [170]. These tactile images can also be applied to the classification of deformable objects [120]. Combining visual and tactile information together also leads to better object recognition [43,160].

Similar to other techniques, tactile-based object recognition has been improved more recently through the use of multi-fingered robotic hands, capable of providing more tactile information about an object than a standard pincer-based robotic end effector [171]. Further, the application of deep learning methods provides more generalised and robust object recognition performance, using simple linear layers [171] and LSTM-based recurrent neural networks [172].

### 4.7. Three-Dimensional Shape Reconstruction

Three-dimensional shape reconstruction enables robotic systems to create accurate representations of objects’ geometries, which can be used to facilitate data collection in complex environments, in-hand object localisation and classification [62]. Tactile sensing contributes significantly to 3D shape reconstruction by providing fine-grained information about the local geometry of the object [173]. Tactile features can be combined with visual features to generate more accurate 3D shape reconstructions [44].

### 4.8. Haptic Feedback

Tactile sensing can be used to provide haptic feedback to a robot operator, allowing them to feel what the robot is touching or grasping. This can improve the operator’s ability to control the robot and perform tasks that require fine motor skills. This capability enhances the operator’s ability to control the robot’s actions, especially for tasks that require fine manipulation, force control, or exploration in unstructured environments [174,175].

Haptic feedback can be achieved by mapping the tactile-sensing data and proprioceptive data (like joint force feedback) from the robot’s end effector to a haptic device, such as a force–feedback joystick or a wearable haptic glove. This enables the operator to perceive contact forces, pressure distribution, surface properties, and vibrations as if they were directly interacting with the object [176,177,178].

### 4.9. Object Pushing

The manipulation of objects through pushing is an essential feature of physical robot interactions. Tactile sensing plays a significant role in object pushing, as it provides valuable information about the contact forces, pressure distribution, surface properties, and frictional interactions between the robot’s end effector and the object.

One of the key challenges in object pushing is to maintain stable contact between the robot’s end effector and the object while applying the desired force. Tactile sensing can be used to monitor the contact state and detect any slipping or unintended contact changes, enabling the robot to adapt its pushing strategy accordingly [29].

Additionally, tactile data can be employed to estimate the frictional properties between the object and the environment, which can be used to predict the object’s motion and plan more effective pushing actions [179]. Using a combination of vision and tactile sensation was shown to produce more accurate object location predictions over extended time horizons [121]. Machine learning techniques have shown promise in enhancing object-pushing capabilities using tactile data. LSTM-based recurrent neural network algorithms have been employed to process and analyse tactile images, enabling the robot to infer object motion, using model predictive control and deep functional predictive control to optimise robot trajectory [29,179].

### 4.10. Summary

We summarise the different key applications of tactile sensors in robotics and automation in Table 2. In this section, we show that there is a wide variety of uses for tactile sensation, ranging from physical robot interaction tasks like grasping and pushing, to feature extraction tasks like texture recognition. Although the provided list and citations are not exhaustive, we aim in this section and the previous one to provide the reader with enough context to understand the following sections on the use of tactile sensation in agri-food and the current issues and shortcomings with tactile-sensing technologies and algorithms that should be addressed.

## 5. Complexities Associated with Tactile Sensors

Despite the recent advances in tactile sensor technology and tactile information-processing techniques, there are still many challenges ahead of the domain for improvement [118,180]. The shortcomings can be categorised into algorithmic and practical challenges. The algorithmic shortcomings relate to the processing techniques of tactile information, which are usually used in three major domains, namely feature extraction, tactile-based robot controllers, and sensory information fusion. The practical complexities include the challenges for calibrating the tactile sensors, the high dimensionality of the tactile information due to the high number of sensing points, and finally the shortcomings related to the hardware integration of tactile sensors into the robotic systems. Figure 8 shows the block diagram of the challenges currently in front of artificial tactile-sensing technology. Knowing the current tactile technology’s shortcomings can provide insightful information for leading the future domain research in the right direction for agri-food applications [118,181,182,183,184]. We will review the shortcomings of the recent research articles presented based on both the algorithmic and practical challenges.

### 5.1. Tactile Feature Extraction

The tactile information acquired from the physical interaction with an object can include a range of different features describing the objects’ (i) geometry, such as size [171,185], edges [186,187], curvatures [188], shape [189,190], or texture [191,192,193,194]; (ii) dynamics, such as inertia [195], stiffness [196,197,198], friction [199,200], deformation [201], or contained fluid [202]; and (iii) properties related to the robot controllers, such as grasping/picking pose [129,203], slippage [134,204], contact point [205], or in-hand object pose [206]. We will focus on the tactile features, which are related to the agri-food domain and present the challenges and shortcomings of the reviewed approaches.

#### 5.1.1. Geometric Features

Based on the variation in size, geometry, and deformability of the fruit and food items, specific hardware and algorithmic requirements are needed for each geometrical feature extraction problem. Zhang et al. [13] combined the force and bending sensors data of a three-finger soft gripper as input to multiple machine learning-based classification models for fruit classification problems. The random forest (RF) classifier results in highest classification scores compared to k-nearest neighbour (KNN), support vector classification (SVC), and naive Bayes (NB) in classifying a fruit set consisting of apple, orange, peach, pear, and tomato. While the fruit object set includes various texture types, there is no large variation in size and geometry. As such, the same tactile sensor and method might not be applicable for detecting a fruit with non-spherical geometry such as a cucumber, or with a larger size such as a melon. Patel et al. [207] utilised the ResNet50 convolutional neural network (CNN) on the tactile images from the Digger Finger sensor to detect different contact geometry classes (circle, hex, square, and triangle) dipped in a granular media (i.e., rice). While the used vision-based tactile sensor and CNN model perform well for the objects with distinguishable edges, it might not work equally well for geometry classification problems for fruits with smoother geometries. Ribeiro et al. [9] used moving average and finite impulse response (FIR) high-pass filter on the cilium-based tactile sensor data for achieving input features in the apple and strawberry smoothness, stiffness, and texture recognition task. The random forest classifier reached 96% and 83% accuracies for apple and strawberry ripeness estimation, respectively. The miniature design of the sensor makes it challenging to be used in agricultural fields in non-laboratory environments. Abderrahmane et al. [208] proposed a CNN model for object recognition capable of generating synthetic tactile features using upconvolutional layers in the model. The BioTac sensor data, besides the first principal components, are used as the input features. Tactile sensor arrays on a soft multi-finger gripper in Zhou et al. [22] help detect the branch interference in apple harvesting using a customised CNN model. The large spatial distribution of the sensors allows successful branch location inference on the fingers. The current approaches are usually limited to specific types of object classes or tactile sensors. Due to the broad spectrum of geometries of food and fruit items, the problem of geometrical feature extraction can be addressed better by the tactile sensors integrated on multi-finger robot grippers, such as in [13,22]. This will allow easier tactile exploration, data collection, and in-field implementation. Feature detection algorithms that can be applied to different tactile data representations (e.g., pressure distribution, tactile image, and force vector), such as machine learning-based models, enable model generalisation to various tactile sensors.

#### 5.1.2. Dynamic Features

Dynamic feature extraction usually demands physical interaction with higher ranges of contact forces compared to geometrical feature extraction. As such, the standard definition of food item damage and bruise [39,40,41] must be passed by the proposed methods. Zhang et al. [4] utilised tactile data for the hardness recognition of a set of fruit and vegetable items. The input features to the KNN and SVM (support vector machine) classifiers are the lower-dimensional vectors achieved by applying principal component analysis (PCA) on raw tactile data for classifying the hardness of apple, kiwi, tangerine, and tomato. Tactile measurements are acquired by squeezing the fruits with a parallel gripper equipped with pressure-based tactile sensors, which can damage softer fruits such as berries type and result in fruit bruises. Scimeca et al. [8] used capacitive tactile sensors on a parallel gripper for estimating the ripeness of mango fruit. The sensor calibration model [209] maps the pressure values to force, and a spring model is used for estimating the stiffness of the fruit. The assumption about the spring model is dependent on the contact geometry and can have variation due to slight changes in the contact state. As such, it needs multiple palpation trials and averaging the estimated stiffness as the final prediction. Although the approach achieved equal classification performance compared to the destructive penetrometer measurements, the heuristic contact model is specific to the fruit type. Blanes et al. [5,12] used six accelerometers’ data attached on the fingers of a pneumatic gripper to classify the firmness of eggplant and mango. The slope and area under the curve of the acceleration signals in the post-grasp phase are used for firmness classification. Although this method can be used in the quasi-static post-grasp phase, where the accelerations are only due to gripping actions, it struggles to differentiate finger grip motion acceleration from the acceleration of the robot hand in the moving phase of a task. Cortés et al. [11] used a similar approach for mango ripeness estimation by a pneumatic gripper equipped with accelerometers and spectrometers. Partial least square regression was used for both accelerations and spectral data for ripeness prediction. Spiers et al. [210] used a random forest classifier for the object class, pose, and stiffness estimation by the single point pressure sensors on the phalanges of the two-finger TakkTile [211] gripper. The object set consists of a small set of objects, and generalisation to novel objects is not explored.

Although tactile-based palpation for dynamic feature extraction could achieve the same level of performance compared to destructive approaches for some fruit items [8,212], they still lack generalisation to different food and fruit types. Reaching human-level dynamic feature extraction requires tactile-enabled soft grippers for non-destructive physical interaction, efficient fusion with visual and proprioception data [213], and processing algorithms with minimal assumptions about the problem features for better generalisation.

#### 5.1.3. Controller Features

Tactile-based robot controllers generate the control actions required for task success based on specific tactile features. For instance, a fruit-picking robot controller can adjust its grasp pose after initial contact according to the more accurate tactile-based pose estimation compared to the initial visual-based estimated pose [130,214]. The latency of the control feature extraction process, consisting of (i) data sampling and (ii) a computation phase, is more critical than geometric or dynamic feature extraction since it determines the reaction time of the robot controllers. Feature detection methods with high computation complexity [215] can increase the system’s latency and reduce the success rate of the controller. The requirement of the latency of the feature extraction method can be determined based on the frequency of the control task.

Hohimer et al. [20] used the *R-C* time delay characteristic of a thermoplastic polyurethane (TPU)-based tactile sensor circuit for detecting the contact on a pneumatic soft gripper. Liu and Adelson [19] performed object orientation estimation by the GelSight Fin Ray sensor. Live orientation estimation is performed by using Poisson reconstruction on a differential tactile image that is calculated by subtracting a reference image (HSV threshold on untouched mode image) from each sensor frame. The precision of the estimation can degrade in dynamic manipulation tasks by faster in-hand object motions. The object set contains artificial fruits, including orange, apple, and strawberry. Zhou et al. [22] used a CNN model for classifying the grasp status in the apple-harvesting task by a four-finger tactile-enabled soft gripper. The grasp status can be one of the following classes: good grasp, null grasp, branch interference, and finger obstructed. The CNN model feeds each finger’s tactile data to separate sub-networks and concatenates the latent features before the output layer. A moving variance method is also used to localise the branch interference by the assumption that branch touch triggers faster pressure variation than apple touch. The grasp status detection and branch interference localisation models reach 87.0% and 83.3% accuracy, respectively. A heuristic classifier is used in [24] on the wrist F/T sensor and the encoders data on mounted finger joints for grasp success estimation in the apple-harvesting problem.Kim et al. [216] used the voltage of a strain gauge-based tactile sensor and motor encoder data for detecting object slip in a pincer gripper. The sensor is not calibrated to measure force or deformation. Wi et al. [201] used estimated contact force and location based on tactile and visual feedback for object deformation estimation. The signed distance functions (SDFs) represent the object deformation in a double-encoder feedforward neural network. The visual feedback may not be available in cluttered scene settings.

The robustness of the feature extraction methods is not tested in various in-field conditions, where there can be large variances in the temperature, humidity, and lighting conditions, and uncertainties, which can significantly influence the tactile measurements. Electronic-based tactile sensors (e.g., capacitive [217] and piezoresistive [218]) have higher sampling rates and are more suitable for real-time robot control but can be occasionally limited to measuring forces only in one direction [219]. Visual-based tactile sensors provide rich information about the contact state for robot control [66,179,220,221] but suffer from low sampling frequency.

### 5.2. Robot Controller

Tactile sensory feedback plays a substantial role in motor control in primates [86,122]. Although visual-based controllers have recently advanced tremendously in robotic manipulation [222], tactile-based controllers are behind the state of the art in visual systems and human-level performance. The main challenges relate to extracting crisp features from high-dimensional tactile data, the complexity of the dynamics of physical contact, and hardware shortcomings, such as having a spatially distributed sensing system that is efficiently integrated with the actuation system similarly to biological sensorimotor loops. We will review the tactile-based controllers which have been used in agri-robotic applications or have the potential to be effectively integrated into an agri-robotic system and discuss the corresponding shortcomings.

#### 5.2.1. Grasp Control

The force closure grip control method has been used as a common approach in robotic grasping and manipulation [223,224,225]. This approach aims to regularise the contact force to a set value either by directly changing the fingers’ position or finger joints’ torques. This approach is difficult to apply to delicate or deformable objects, as the grasp width and tactile force can change with object deformation. He et al. [226] used a force closure approach on a gripper with soft finger pads with air cavities to avoid damaging delicate objects. Wen et al. [227] introduced a high-precision grip force control of delicate objects, where the grip force is regulated based on tactile feedback and a human teleoperator hand’s electromyography (EMG) signals. These approaches require an object model to adjust the grip force or width. Yin et al. [228] leveraged deep reinforcement learning for in-hand object rotation control based on tactile feedback, which is a data-driven approach. The tactile data include sixteen contact sensors on the palm and phalanges of the Alegro hand. The deep RL method takes tactile and robot proprioception data as input and generates future joint torques to achieve a desired object rotation. The reward function is defined for the rotation task, and having a global reward function for in-hand manipulation can be intractable to find. Slip avoidance controllers regulate the grip force to prevent future slip incidents [138,139,140,229,230]. Achieving human-level dexterity requires an efficient integration of tactile sensors and actuation systems in an anthropomorphic gripper. The active touch in humans [231] generates exploratory actions to achieve suitable tactile feedback for further adjustment of the grip force and pose. Nonetheless, customised grip control for handling certain types of food and fruit items can be more cost efficient than designing an expensive human-level hand for universal manipulation applications.

#### 5.2.2. Motion Planning

Proprioception sensory feedback helps primates to not only adjust their grip force but optimise the motion of different body parts, such as the hand and the arm [232]. The bulk of the robotic literature seeks to exploit tactile feedback primarily in grip control and disregards hand and arm motion planning based on tactile feedback. Schuetz et al. [28] proposed an obstacle avoidance motion planning framework based on tactile feedback for an agriculture robotic manipulator. Using feedback linearisation and gradient-based optimisation with an objective function consisting of collision avoidance and joint limit terms, the motion planning approach avoids obstacles in the task space. The proposed approach is applied for the collision of one link of the manipulator. Multiple link collision detection and prevention can make the control problem untraceable. An online robot trajectory optimisation approach was proposed by [150] for object slip avoidance. The objective function consists of (i) the distance from a desired pre-planned reference trajectory (e.g., minimum time and minimum jerk), and (ii) future slip likelihood in a horizon. The optimisation helps the motion-planning pipeline to maintain both the reference trajectory behaviour and avoid object slip simultaneously. Nazari et al. [29] introduced tactile deep functional predictive control in strawberry-pushing tasks. The proposed data-driven controller adjusts the robot’s Cartesian velocity to prevent losing contact with the strawberry stem during the pushing task execution based on the sensory feedback from a camera-based tactile finger. Off-the-shelf motion planning libraries [233,234] lack integrating tactile feedback in problem formulation, such as the approaches in [28,29,150] for more robust closed-loop motion planning.

#### 5.2.3. Learning from Demonstration

Learning from demonstration (LfD) is a data-driven control approach, where the robot controller can be learned from expert demonstration [235]. This approach is especially preferred in scenarios, where the control problem cannot be formulated analytically. Misimi et al. [27] proposed a LfD method for the compliant grasping of deformable food objects, which can automatically discard inconsistent demonstrations. The method combines visual and tactile feedback for the robust autonomous grasping of food items. A breast cancer examination robot controller was studied in [236] by a novel LfD technique using deep probabilistic movement primitives. The human demonstration data consist of reach to palpate and palpation trajectories, and the model can generalise to unseen breast poses. Tactile-based LfD has a high potential to improve the robotic harvesting systems using the demonstration of expert human pickers. Recent research in [237] tried to recognise human fruit-pickers’ activities during avocado fruit harvesting. DelPreto et al. [35] collected a multi-modal dataset of human activities in a kitchen environment, including tactile sensors on a human subject’s hand. These types of data and studies can be used for the automation process of applications, such as harvesting and kitchen robotic systems by using LfD approaches. The main challenge ahead of tactile-based LfD methods is the level of generalisation to novel tasks and test conditions.

### 5.3. Sensor Fusion

Combining the sense of touch with other sensing modalities, such as vision and audition, helps humans to have a robust and intelligent multi-modal perception system. Nevertheless, robotic systems fall far behind in finding an efficient universal fusion approach for multi-modal sensing. Recent advances in deep learning leveraged some techniques for effective sensory data fusion [238,239]. Dong et al. [3] proposed a robotic visual–tactile perception learning based on an auto-encoder neural network which consists of a modality-specific knowledge library and modality-invariant sparse constraint to learn both intra-modality and cross-modality knowledge. Each sub-network of the model requires retraining for every new task, which is time and cost expensive to use the model in different multi-modal perception problems. Misimi et al. [27] generated the initial grasp pose by the visual data and used tactile feedback for the final adjustment of the grip pose and force after touching the objects. Wi et al. [201] combined point-cloud data and tactile feedback in an object-conditioned feedforward neural network for estimating the deformation of the object in hand. Luo et al. [162] combined visual and tactile data in a CNN model for texture recognition. Calandra et al. [240] used a similar approach of visuo-tactile fusion by a CNN for grasp stability prediction. Mandil and Ghalamzan-E [121] proposed a multi-modal video prediction model by combining tactile and visual data in novel action-conditioned recurrent neural networks. A strong potential of visual and tactile fusion, which is not yet well explored, lies in the problem of fruit ripeness estimation, where the visual data capture the colour features for ripeness and tactile data measure the stiffness of the fruit. Proximity sensors data are also combined with tactile data in applications such as surface crack detection [241] and safety control in human–robot interaction [242], which can be transferred to the agri-robotic domain for safer human–robot interaction.

### 5.4. Sensor Calibration

Calibrating the tactile sensors to map the raw sensory reading, such as resistance, capacitance, voltage, current, length, light intensity, magnetic field, image, or vibration to force values can be very challenging based on the raw sensory data type and dimensionality. The common approach is to fix either the tactile sensor or a load cell (force sensor), and apply precise constant forces to the fixed object by the other to record both the tactile sensor’s raw reading and the load cell’s force values. After a data collection phase of applying forces of various magnitudes and locations, a regression model will be trained for sensor calibration. Depending on the complexity of the regression task, the regression model’s variance and bias can change. Wang et al. [243] used moving least square (MLS) for tri-axial force calibration from the magnetic field values. The calibration test setup is shown in Figure 9a. Yuan et al. [173] used a CNN model to map Gelsight’s markers pattern motion to force values. The entropy of the marker displacement field is used for slip calibration. Khamis et al. [244] used a camera with a high frame rate to measure the sensor taxel’s deformation to further map to normal and shear force values and also for slip calibration (see Figure 9c). Scimeca et al. [8] calibrated a capacitive tactile sensor on a parallel gripper by pinching a metal cuboid that has a force sensor attached to one of its faces. In the calibration procedure, one finger is fixed and the second finger performs a linear motion with constant speed while pressing the force sensor until the maximum displacement of the taxels reaches a threshold. This method is limited to normal force calibration and cannot be applied for calibrating shear forces. Furthermore, the logarithmic function used for mapping pressure to force is specific to the capacitive tactile sensor and cannot be used for other sensors with different hardware. A similar setup was created in [245] for calibrating a photoelastic haptic finger mounted on the Franka Emika robot hand. The calibration setup is shown in Figure 9b. Bio-inspired cilium-based tactile sensors can detect very small-scale touch features [9]. However, due to the miniature size of the hairlike sensor structure, it is challenging to calibrate the sensor for measuring force values. Figure 9 shows the tactile sensor calibration setup for the reviewed research items. Knowledge transfer for sensor calibration could save time and cost for performing the calibration procedure and physical interaction data collection for every new tactile sensor.

### 5.5. The Curse of Dimensionality

The spatial distribution of tactile sensing over a large area and having multiple sensing points (taxels) make the dimensionality of tactile information very high. High-dimensional sensory data are challenging to deal with from the perspective of both control and feature extraction. As such, dimensionality reduction by deep neural networks is widely used for tactile data processing in various tasks, such as slip classification [247], object pose estimation [248,249,250], and grasp stability prediction [147,251]. Zhang et al. [4] applied PCA on tactile data to achieve compact features for KNN and SVM classifiers for fruit hardness recognition. Dong et al. [3] used the VGG-16 model on both visual and tactile images to achieve a vector input to a multi-modal auto-encoder in a fabric classification task. Funabashi et al. [252] compared the performance of DNNs (deep neural networks), CNNs (convolutional neural networks), and RNNs (recurrent neural networks) models for the object recognition task using sixteen uSkin tactile sensor arrays integrated on the Allegro hand. Abderrahmane et al. [208] combined the BioTac tactile data with the first four principal components obtained by PCA for the object recognition task. The major shortcoming of the existing approaches is that data compression methods are used for specific tasks and features, whereas in the human somatosensory system, the encoding of tactile data happens universally for all tactile features [253,254].

### 5.6. Hardware Integration and Scalibility

Each fingertip of the human hand contains 3000 tactile mechanoreceptors, which are, on average, 0.4 mm away from each other [255]. Reaching the same level of measurement and spatial resolution for artificial tactile sensors is very challenging with the current hardware technology. As such, integrating tactile sensors in small areas of robot fingers and gripper is a challenging task. Crosstalk between sensing points [37], the required space for wiring [36], and the size of single sensing hardware are the main bottlenecks to increasing the spatial resolution of tactile sensors. A desired feature for integrating the sense of touch to robotic systems is to have a tactile-enabled hand that has efficient integration of tactile sensing and actuation in both the hardware and software sides. In this section, we review some of the research work that proposed tactile sensors which are integrated within a robotic system, including robotic grippers, soft grippers, and the universal attachment of tactile sensors over a robot body.

#### 5.6.1. Rigid Links Grippers

Abdeetedal and Kermani [256] proposed an underactuated two-degrees-of-freedom gripper, which has integrated load cells within each finger’s phalanges and a potentiometer on the finger joints. Finite element analysis is used to optimise the location of the load cells with the phalanges and contact point estimation is implemented using the force and torque feedback. The grasp controller has two gripping options, namely, the precision and power grasp of the fingers. Zhang et al. [13] used flexible silicone sheets for the fingers in a multi-finger gripper with integrated force and bending sensors. The effectiveness of the flexible fingers is tested in the grasping of different types of soft and hard fruits. Cook et al. [18] used nine capacitive tactile sensors alongside temperature detectors in the phalanges of a tri-gripper. The paper demonstrated a pick-and-place task for various fruits. However, the pick-and-place pipeline is fully working on visual feedback and does not use tactile data. Ntagios et al. [257] used soft capacitive pressure sensors on the distal phalanges of a 5-finger gripper. The pressure sensors measure one value for the whole contact surface, which lacks having a distribution of pressure over the contact area. This means a low spatial resolution for the finger-integrated tactile sensor. Piezo-capacitive tactile sensors with conformal microstructure were proposed in [258] and integrated into a multi-finger robot hand for Braille and roughness detection. The response speed is 25 ms, which cannot be sufficient for dynamic manipulation tasks. Fabricating an array of sensors with this technology could further decrease the response speed, which shows the limitation for expanding the spatial size of the sensor similar to biological skin. Dischinger et al. [24] embedded 2 × 2 pressure sensor arrays below the finger pads of a three-finger gripper for apple harvesting. The tactile-enabled gripper was tested in the field for branch interference detection. The IMU and finger encoder data from finger pads and joints, respectively, were preferred over the tactile data for grasp model estimation. Branch interference made the tactile data unreliable for detecting a successful grasp or object slip. No bruise tests were conducted in the real-world apple harvesting tests. Blanes et al. [12] used accelerometer data on a pneumatic robotic gripper for fruit firmness estimation. Funabashi et al. [252] integrated sixteen uSkin tactile sensors on Allegro hand palm, phalanges, and finger pads. A gripper that has good integration of the sensing of (I) tactile data on fingers and palm, (II) joint angles of each finger joint, (III) and joint torques on each finger joint is currently missing in the literature.

#### 5.6.2. Soft Grippers

Soft grippers can be more suited to the application of agri-food manipulation compared to hard grippers based on their adaptive soft shape and lower actuation forces [259]. Tactile-enabled soft grippers can leverage bruise-free fruit harvesting with sufficiently good tactile perception. An adaptive compliant gripper was proposed by Liu and Adelson [19] for compliant grasp control using the GelSight Fin Ray sensor. A silicone gel pad is attached to a printed deformable finger, and internal illumination helps the camera sitting in the finger base measure the displacement of the patterns on the pad. Tactile images are used for in-hand object orientation estimation. Flexible thermoplastic polyurethane (TPU) material inspired the fabrication of a range of tactile sensors working on capacitive and piezoresistive fundamentals [20]. The sensor is integrated into a pneumatic-based soft gripper for apple harvesting. Contact force localisation is challenging for TPU-based tactile sensors. Zhou et al. [21,22] integrated twenty-four piezoresistive tactile sensors (RX-M0404S) in a four-finger soft gripper for apple harvesting. Each finger contains six tactile arrays embedded by a thin silicone skin, and each tactile sensor has a 4 × 4 taxel configuration. The tactile-enabled gripper is integrated into a UR5 robot, which is mounted on a mobile platform, and the system is extensively tested in the field for apple harvesting. He et al. [226] introduced a gripper with soft fingertips containing a cavity inside to measure the applied pressure. The spatial resolution of the proposed method for pressure sensing can be extremely low in the case of scaling up the sensors over larger areas.

#### 5.6.3. Tactile Skin

Although the primary focus on tactile sensor integration in the robotic systems has been on robot manipulators’ hands, developing an electronic skin over non-hand areas has been partially investigated [260]. Schuetz et al. [28] embedded a tactile sensor in one of the middle links of an agriculture robotic manipulator. The tactile sensor has two rigid frames, which are connected to each other by four force sensors. The force values are used to achieve the applied force and torque on the robot link utilised by the obstacle avoidance controller. The contact location cannot be localised on the arm, and a middle point on the link is assumed as the default contact point. Patel et al. [207] introduced Digger Finger with a GelSight-type visual tactile sensor restructured in a cylindrical architecture for inspecting hard objects in granular environments. The test tasks include inspecting metal objects in rice. Zhang et al. [261] introduced a resistive-based large tactile skin used to cover the links of the UR5 robot. The tactile skin has a divided texture, where each part can measure the normal pressure. The developed tactile skins are usually incapable of measuring shear forces and have a small spatial resolution.

## 6. Future Trends and Conclusions

In this review, we explore the use of tactile sensation in agri-food. We provide an overview of tactile sensation hardware and its general uses in robotics and sensing. We discuss in depth the use of tactile sensation in agri-food and the current shortcomings of tactile sensation technologies. Tactile sensation has been employed in three primary areasof agri-food research: First is the design of robotic harvesting systems that use tactile sensors to harvest foods delicately and assess the ripeness and quality of fruits and vegetables based on factors such as firmness, texture, and other attributes. Second, the incorporation of tactile sensing in automated packaging and handling systems enables the gentle and accurate manipulation of delicate produce to reduce damage and food waste. Tactile sensation is starting to emerge in kitchen robotics, such as in food-pouring techniques and cutting processes. However, the research in this area is still in its early stages, offering ample opportunities for further exploration and development.

Tactile sensing has the potential to contribute significantly to a broader array of agri-food applications. One such application includes the monitoring and maintenance of livestock health by facilitating the early detection of injuries or diseases through the examination of animal coats, skin conditions, and overall body condition. Furthermore, the application of tactile sensing, although primarily employed in food quality evaluation, can also be leveraged for monitoring plant health and detecting diseases. This is attributed to the sensor’s capability to perceive the physical properties of plants, which may act as indicators of their overall health. Additionally, extending this approach towards pest control and identification might further broaden its scope. The use of tactile sensors could contribute additional features required for detecting and identifying a wide array of pests related to agri-food products. This might be possible by interpreting physical interactions with the crops. For instance, detecting alterations in the plant’s physical structure due to pest infestations or diseases could be achieved through tactile features and not through remote-sensing units, thereby offering a promising direction for future research. In the long run, ongoing research in these areas will lay the groundwork for advanced applications of tactile sensing in agri-food, promoting sustainable agriculture and improved food security.

The hardware development of artificial tactile sensors is still facing scalability challenges for having a large number of sensitive high-spatial-resolution taxels over large areas. The recently proposed self-powered tactile skin partially addressed the scalability problem, but it still lacks high dynamic ranges and has a low resolution for large contact forces. Improving the dynamic range of self-powered tactile sensors can introduce the next generation of artificial skin suitable for various applications, including the agri-food domain. There is no standard definition of wear and tear testing of tactile sensors to measure their endurance in long-term real-world applications such as harvesting in agricultural fields. Future research can benchmark wear testing of the tactile sensors for easier endurance comparison between different tactile sensors in the robotic community. Most of the proposed tactile-sensing technology have complicated calibration and integration procedures, which can limit their applications to research environments. As such, future work can explore modular design with easier integration with off-the-shelf hardware systems and a unified tactile feature extraction approach, which simplifies the sensor calibration. The commercially available tactile sensors are currently very expensive to afford, and future large-scale commercialisation requires significant optimisation of the production process.

In conclusion, this review delves into the potential of tactile sensation in the agri-food sector, highlighting its primary applications in robotic harvesting systems, automated handling, and emerging uses in kitchen robotics. Tactile-sensing technology also holds promise in livestock health monitoring and plant health assessment, which could further revolutionize sustainable agriculture and enhance food security. However, challenges remain in scaling up the hardware, improving dynamic ranges, and increasing resolution for large contact forces. The development of self-powered tactile skin has enabled progress in addressing some of these issues, but further advancements are required. To facilitate the adoption of tactile-sensing technology, future research should focus on establishing standardized wear-and-tear testing, creating modular designs for seamless integration, and simplifying calibration procedures. Lastly, making commercially available tactile sensors more affordable through optimized production processes will be essential for widespread implementation in the agri-food industry.

## Figures and Tables

**Figure 1 sensors-23-07362-f001:**
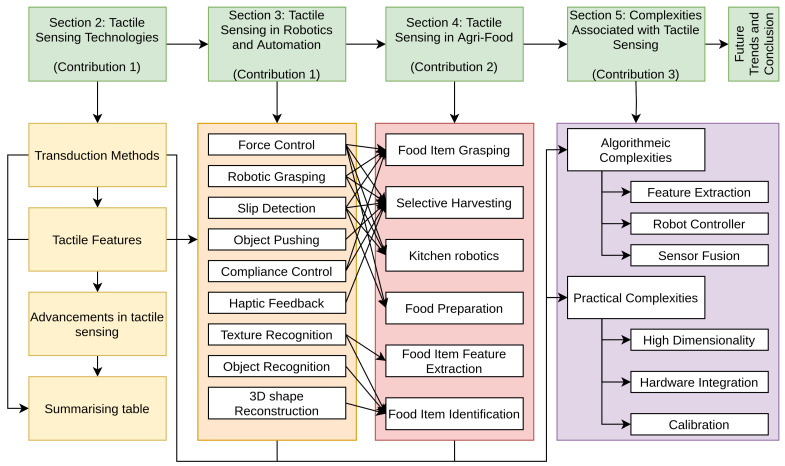
A summary of the work and contributions in this review. We introduce tactile sensing technologies and their general applications/algorithms in Section 2 and Section 3. This context is then used to go in depth into the current use of tactile sensation in the agri-food sector in Section 4. Finally, we provide a detailed analysis of the current tactile-sensing technology shortcomings.

**Figure 3 sensors-23-07362-f003:**
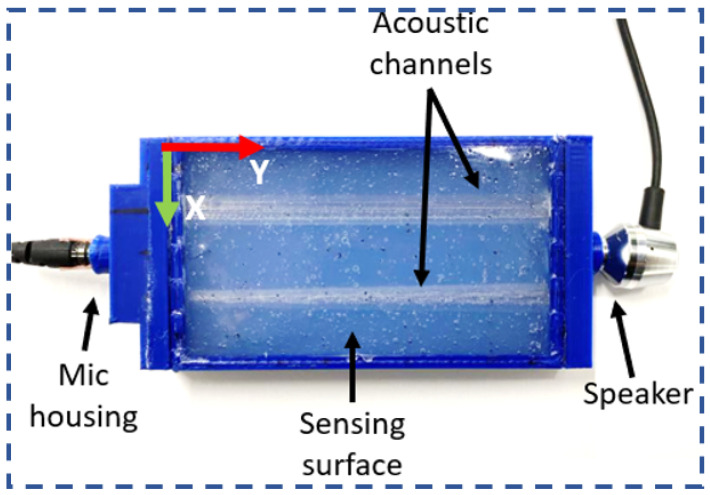
Example of low-cost tactile sensing: Sensorizing a flexible soft skin by constructing internal acoustic channels.

**Figure 4 sensors-23-07362-f004:**
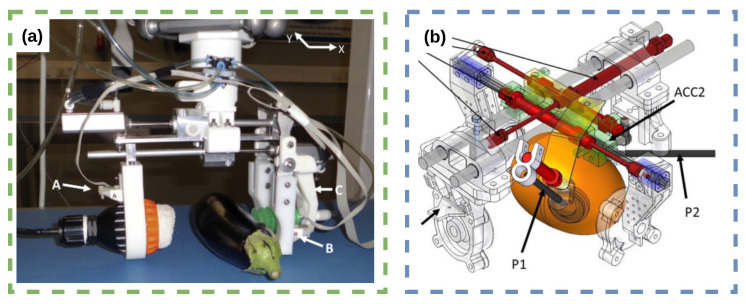
Examples of tactile sensation systems used to extract features from food items. (**a**) A pneumatic robot gripper for sorting eggplants by firmness Blanes et al. [5], (**b**) mango ripeness assessment through visuo-tactile system Cortés et al. [11].

**Figure 5 sensors-23-07362-f005:**
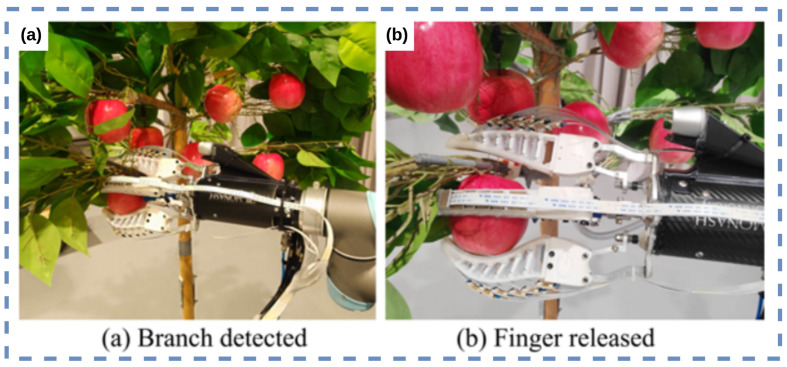
Novel multi-fingered, soft harvesting end effector for apples (**a**,**b**) novel ability to remove grasping finger in situations of physical obstruction Zhou et al. [22].

**Figure 6 sensors-23-07362-f006:**
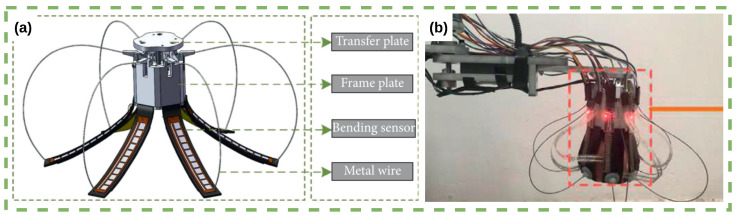
Soft tactile-sensing end effector that measures contact force distribution and finger deformation during grasping for food item identification Zhang et al. [13] (**a**) shows the schematic of the tactile system and (**b**) shows the real tactile system grasping a soft fruit for identification.

**Figure 7 sensors-23-07362-f007:**
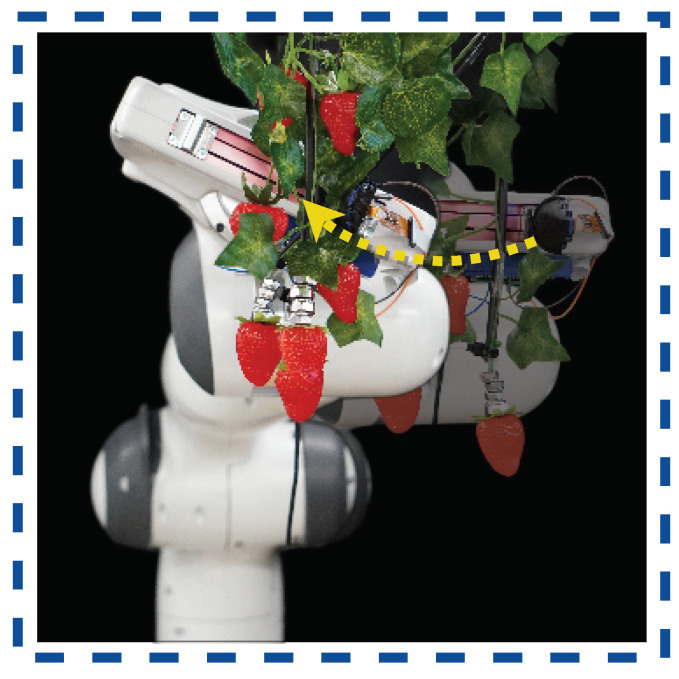
Physical manipulation of harvesting environments using tactile sensation: strawberry cluster manipulation using visual tactile sensor.

**Figure 8 sensors-23-07362-f008:**
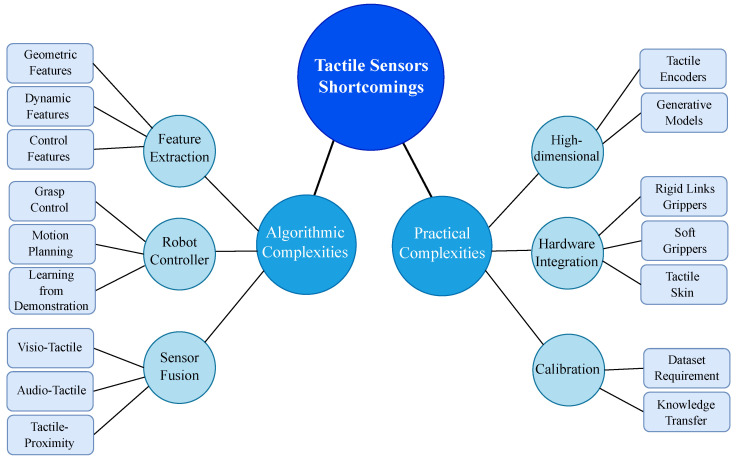
The block diagram of challenges of using tactile sensors.

**Figure 9 sensors-23-07362-f009:**
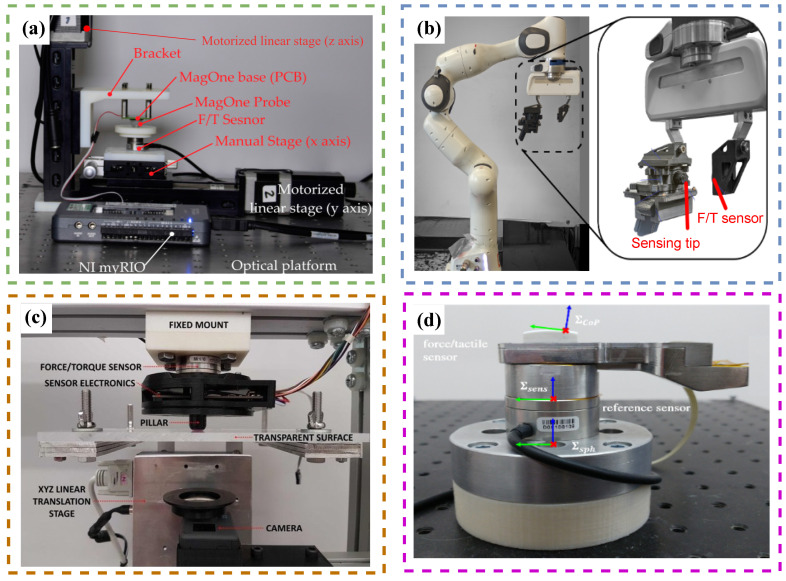
Examples of tactile sensor calibration setup. (**a**) Force calibration of a magnetic field-based tri-axial tactile sensor [243], (**b**) force calibration of a photo-elastic tactile sensor [245], (**c**) force and slip calibration of an optical-based tactile sensor (papillarrary) tactile sensor [244], and (**d**) force calibration of an opto-electronic based tactile sensor [246].

**Table 2 sensors-23-07362-t002:** Summarising the most important uses of tactile sensors in robotics and automation.

Robot Task Type	Tactile Sensor Type	Cited Research
Robot Control Tasks	Force Control	[45,86,122,123,124,124]
	Robotic Grasping	[123,125,126,127,128,129,130,131,132,133]
	Slip Detection	[123,134,135,136,137,138,139,140,141,142,143,144,145,146,147,148,149,150]
	Object Pushing	[29,121,179]
	Compliance Control	[163,164,165,166,167]
	Haptic Feedback	[174,175,176,177,178]
Feature Extraction Tasks	Texture Recognition	[47,60,151,152,153,155,156,157,158,159,160,161,162]
	Object Recognition	[43,45,86,120,156,160,168,169,170,171,172]
	3D Shape Reconstruction	[44,62,173]

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
