# Peer review of "Tactile-Sensing Technologies: Trends, Challenges and Outlook in Agri-Food Manipulation"

_sensors, 2023, doi:10.3390/s23177362_

Round 1
Reviewer 1 Report
The authors provided an extensive tactile internet review.
The introduction stated the problem and the motivation of this work clearly. I would have liked to see a schematic figure relating contributions to sections and how the sections are related to each other.
The section two provided a brief analysis of the current type of sensors available for tactile application. However, the fiber Bragg grating (FBG) was not described in this part, but it was announced on pag. 16 line 603. Why this kind of sensors were not introduced in the section two?
The section three provided an analysis of tactile sensors applications. I would have liked to see a table summarizing the most important aspect of tactile sensors applications.
The section four provided an analysis of tactile sensors applications in agro-food. I would have liked to see a review about tactile sensors applications to detect pests. Also, the best wait to cite or make a reference to others word is to states the first author surname instead of using the reference. For example:
In paragraph line 477: "[11] introduced a novel robotic...". Please avoid this. In the same paragraph line 481: "Similarly, Blanes et al. in [12] presented pneumatic..." this is a correct way to reference. Please fix it.
Another relevant topic in robots is the Degrees of Freedom DoF and I did not find anything that explain about it. I just found the acronym in line 614 without a clear explanation of it.
The section five provided an analysis of current challenges. In some subsection was difficult to identified the open challenges. Each subsection looks like a review extension of the previous section instead of stating the open challenges clearly.
To sum up, the survery is impresive, but it can be improved.
There are some typos that need attention:
- line 932: Scalibility
- line 545 ot=f
- line 111 utlises
Also, there are some acronyms that have not been defined in the text, for example:
- line 614 DOF
- line 723 SVM
- line 928 DNN, RNN
- lines 339 LSTM.
And among others. Please review of the acronyms.
The quality of english languange is high. However, the authors used some words that are not common. This can difficult to read and understand the text to people that are no native and with an average english level. My recomendation is to use more general words and use synonims.
Reviewer 2 Report
The paper review about the tactile sensors for Agri-Food robotics. We believe that this is a very important field of application for tactile sensors and contains important content that will attract attention in the future.
However, there are several typos and possibly inaccurate context regarding this review article, which need to be corrected.
(1) Section 2.1.1. is supposed to discuss resistive andor Piezoresistive sensors, but the last line of the section is about Piezorelectric sensors, and there is no comparison with Resistive and or piezoresistive sensors. Thus the sentenses is not appropriate to write in this section.
(2) Regarding Fig. 1, there are structural diagrams for other sensors, but there is no diagram for piezoelectric sensors. I guess it is important to add figure about piezoelectric sensors.
(3) Regarding Fig. 1.(d), a diagram of the EIT sensor is included, but unlike the others, it is an external view of the sensor and does not show detail of the sensor.
(4) In lines 220, 224, and 227, the Celsius temperature symbol is probably changed in 0C.
(5) In section 2.3.1, a tactile sensor that measures the state of the contact area using ultrasonic waves is introduced as an unconventional sensor, but in 1995, Shinoda et al. published a paper on a similar sensor in IEEE Control Systems Magazine vol. 15, Issue 1, titled "Ultrasonic Emiision Tactile Sensing". This group had also published some papers about similar sensors, thus it seems unnatural to say that this principle as "Unconventional".
(6) In line 247 there is miss indentation.
(7) Line 269, period is missing.
(8) Line 395, "gas been", which seems to be a typo for "has been".
(9) Line 443, the link to the cited document is broken.
(10) Section 5.1.3, line 790, it is stated that piezoresistive sensors generally measure only normal force. However, as the authors also cite in their reference 124, many of today's piezoresistive sensors, especially MEMS sensors, are reportedly capable of measuring multi-axial forces. This sentence seems unnatural to the trend.
In particular, there are many typos in the document, and we believe it is essential to check the document carefully.
